# Structure of ATTRv-F64S fibrils isolated from skin tissue of a living patient

Jun Yu [1,4], Xuefeng Zhang [1,4], Sandra Pinton[2], Elena Vacchi [2], Andrea Cavalli [3], Matteo Pecoraro [3], Giorgia Melli [2] & Andreas Boland [1]

Amyloid transthyretin-derived (ATTR) amyloidosis is a degenerative, systemic disease characterized by transthyretin fibril deposition in organs like the heart, kidneys, liver, and skin. In this study, we report the cryo-EM structure of transthyretin fibrils isolated from skin tissue of a living patient carrying a rare genetic mutation (ATTRv F64S). The structure adopts a highly conserved fold previously observed in other ATTR fibrils from various tissues or different genetic variants. Mass spectrometry was used to evaluate fibril content and to identify common post-translational modifications. The structural consistency between ATTR filaments from different tissues or patients validates non-invasive skin biopsy as a diagnostic tool.

Amyloid transthyretin-derived (ATTR) amyloidosis is one of the most prevalent forms of systemic amyloidosis and encompasses two types: a genetic (ATTRv) and a sporadic wild-type (ATTRwt) form. ATTRv amyloidosis arises from pathological mutations in the *TTR* gene, including amino acid substitutions, duplications, and deletions[1]. Transthyretin (TTR) mutations often destabilize the native TTR fold, leading to amyloid formation in multiple organs such as the heart, kidneys, liver or skin[1,2]. To date, a total of 216 mutations have been identified, including 200 amyloidogenic and 16 non-amyloidogenic mutations[1,3]. The clinical manifestations of hereditary ATTRv amyloidosis are highly variable, however, the predominant forms are characterized by peripheral polyneuropathy and an early-onset disease[4]. In contrast, ATTRwt amyloidosis is associated with aging-related factors or unknown processes that lead to the extracellular deposition of ATTR fibrils in tissues[5], and late-onset cardiomyopathy is typical[6].

Based on their composition and morphology, ATTR fibrils are classified into two main types. Type A fibrils are composed of full-length (127 amino acids) or fragmented TTR molecules and can be found in ATTRwt and most ATTRv variants. In contrast, type B fibrils only contain full-length TTR[7,8]. Importantly, different fibril types are associated with specific diseases in neurodegenerative pathologies[9]. In addition to genetic mutations, post-translational modifications (PTMs)

can influence the formation and characteristics of amyloid fibrils. PTMs have been implicated in altered behaviour of many amyloid proteins, including amyloid β, tau, α-synuclein, huntingtin, and TDP-43[10]. Several types of modification, including phosphorylation[11], acetylation[12] and ubiquitination[13] have been described as modulators of aggregation rate or extent, aggregate stability, and cytotoxicity.

A growing number of cryo-EM structures of ATTR fibrils, extracted from post-mortem tissues such as the heart, eyes and nerves provided critical insights into the structural organisation of ATTR amyloid fibrils[14–22]. All structures share a common, relatively compact and β-sheet-rich fold that has been described as spearhead-shaped[14]. Despite their overall structural homogeneity, ATTR fibrils from different tissues exhibit local variations in a region that spans amino acids G57 to G67[19], referred to as 'gate' region. For example, in cardiac fibrils of ATTRv-I84S patients four distinct gate states have been observed, named open, closed, broken and absent. In contrast, in ATTRv-V30M fibrils from the eye only one gate type was observed, called blocking gate[15,19]. ATTRwt cardiac fibrils from five patients and ATTRv fibrils (V20I, P24S, V30M, G47E, T60A, V122I and V122Δ) from various tissues show a closed gate near a polar channel. Polymorphism has also been observed in the number of protofilaments. Most ATTR fibrils consist of a single protofilament under cryo-EM conditions, with the exceptions of ATTRv-V122Δ cardiac fibrils that contain one or two protofilaments

---

[1]Department of Molecular and Cellular Biology, University of Geneva, Geneva, Switzerland. [2]Institute for Translational Research (IRT), Faculty of Biomedical Sciences, Università della Svizzera italiana (USI) and Ente Ospedaliero Cantonale (EOC), Bellinzona, Switzerland. [3]Institute for Research in Biomedicine (IRB), Università della Svizzera Italiana (USI), Bellinzona, Switzerland. [4]These authors contributed equally: Jun Yu, Xuefeng Zhang. ✉e-mail: Giorgia.Melli@eoc.ch; Andreas.Boland@unige.ch

and ATTRv-V30M fibrils from the eye that are formed by multiple protofilaments[15,22].

Recently, we showed that skin biopsy is an extremely sensitive, minimally invasive test for detecting and typing ATTR amyloidosis[23]. In this study, we use immunohistochemistry, mass spectrometry, and cryogenic electron microscopy (cryo-EM) to describe the molecular composition and the structural characteristics of amyloid fibrils extracted from ankle and thigh tissues from skin biopsies of a living ATTRv-F64S patient. The ATTR fibril structure of this genetic variant has not been determined yet. Our work reveals that ATTRv-F64S fibrils contain one protofilament and, less frequently, two protofilaments. The single protofilament adopts a near-identical fold to that of ATTRwt and most ATTRv fibrils, featuring a closed gate. Our structure of ATTR fibrils derived from a skin biopsy of a living patient demonstrates that sufficient quantities of amyloid fibrils can be extracted from minimal amount of skin tissue (between 5-10 milligrams). The structural conservation of ATTR fibrils across various tissues, including skin, further corroborates skin biopsy as a minimally invasive test for detecting, typing and determining the structure of ATTR amyloid fibrils.

## Results

### Characterisation of ATTR fibrils from ankle and thigh skin biopsies of a living patient

In a first step we quantified the abundance of ATTR amyloid fibrils in two different skin sections, namely from ankle and thigh tissue. Amyloid deposits were found in the subepidermal layers and dermis, primarily around arterioles and sweat glands, consistent with our previous report[23]. Filaments were stained with Congo red dye that results in red or pink deposits that can be observed by brightfield (BF) microscopy (Supplementary Fig. 1a, BF). Using polarized light (PL) the characteristic birefringence of amyloid fibrils was detected in both tissues (Supplementary Fig. 1a, PL). When comparing ankle and thigh tissue, a stronger Congo red staining and higher birefringence was detected in ankle tissue from this patient. PGP9.5 (Protein Gene Product 9.5) antibody staining showed reduced intraepidermal nerve fibre density in both samples, confirming small fibre neuropathy (Supplementary Fig. 1a, PGP9.5).

We next extracted ATTR fibrils from ankle and thigh tissue obtained by skin biopsy. Fibril extraction and purification was performed using a water extraction protocol previously shown to preserve the structure of the amyloid fibrils[24]. The purity and quantity of isolated fibrils from the two tissue samples was assessed by silver-stained SDS-PAGE gels. The gels showed two or more bands running at approximately 12-15 kDa, indicating the presence of full-length (127 amino acids) and fragmented TTR. Consistent with our quantification in tissue sections, we observed much stronger intensities of TTR monomer bands in the sample isolated from ankle compared to thigh tissue (Supplementary Fig. 1b, dashed box). A higher abundance of fibrils was also detected in ankle tissue versus thigh tissue using negative-stain microscopy (Supplementary Fig. 1c).

### Characterisation of ATTRv-F64S fibrils by mass spectrometry

Next, minute quantities of fibrils from ankle and thigh tissue (300 ng and 30 ng, respectively) were analysed using bottom-up liquid chromatography-tandem mass spectrometry (LC-MS/MS) with high-sensitivity acquisition on a trapped ion mobility (TIMS) mass spectrometer. TTR was readily identified in both samples with 20 and 22 unique peptides, respectively. This resulted in a near full coverage (92.9%) of the TTR mature form with only the first nine amino acids missing (Supplementary Fig. 2a, b and Supplementary File 1). An N-terminal free semi-specific search further increased the coverage to a maximum of 98.4%, including three additional peptides that further confirmed the presence of full-length TTR within the fibrils (Supplementary Fig. 2c, d and Supplementary File 1). For peptides encompassing the F64S mutation site, both wild-type and mutant sequences

were detected. When analysing their relative intensities, we consistently observed higher intensities for peptides carrying the F64S mutation compared to wild-type peptides (Fig. 1a–c). Because the mutation could in principle alter the ionization properties of the peptides, we also analysed a piece of ankle skin biopsy without performing fibril extraction, to inspect bulk epidermal TTR by LC-MS/MS. In this setting, we indeed observed similar intensity levels of wild-type and mutant peptides, in line with the patient's heterozygosity for the TTR-F64S mutation (Supplementary Fig. 3 and Supplementary File 1). Our results suggest a higher fraction of mutant versus wild-type TTR in ATTRv-F64S fibrils, consistent with a destabilisation effect of the native TTR fold by this mutation, eventually leading to amyloid formation. We then examined the fibril data for post-translational modifications, including acetylation, methylation, phosphorylation and ubiquitination. Using a deep learning prediction module of the MSFragger search tool, we confidently identified fourteen PTMs on nine sites. These sites were filtered for their localization probability and spectral similarity to predicted MS/MS spectra above 80% (Supplementary Table 1). Eight of the PTM sites were present in fibrils extracted from ankle and thigh tissues (ten hits if the similarity score is reduced to 79%), and were consistently found in independent scans, with a maximum of 21 spectral matches. K15 ubiquitination was the only PTM identified from a single spectrum, however, this modification has also been described in the PhosphoSitePlus database[25]. Other previously identified modifications that we corroborate in this study include K15 acetylation (enriched in the ankle tissue from this patient), S52 phosphorylation and Y105 phosphorylation[25–27]. Three PTMs were identified in peptides that include the F64 mutation site. Of these, T49 phosphorylation and K70 acetylation were specifically assigned to F64S-TTR, while S52 was found in the wild-type and mutated TTR form. These results show that comprehensive and reliable PTM analysis can be performed on ATTR fibrils extracted from skin biopsies in an unbiased manner without prior enrichment for modified peptides.

### Cryo-EM structure of ATTRv-F64S fibrils

ATTR fibrils obtained from roughly 5-10 mg of ankle tissue allowed the preparation of a total of four EM grids. Based on visual inspection of the micrographs, the imaged fibrils appeared largely uniform in morphology, although aberrant and low-abundance ATTR fibrils may also exist (Fig. 1d and Supplementary Fig. 1c). Automated filament picking was performed using a modified Topaz pipeline[28], followed by helical reconstruction in RELION[29]. Two-dimensional (2D) class averages revealed a spacing of ~4.8 Å, as previously described[14] and showed two types of fibrils: a single twisted protofilament and a twisted dimer (Supplementary Figs. 4a, b). Three-dimensional (3D) classification of single protofilament segments yielded one class with high-resolution features (Fig. 1e, f). A subsequent 3D reconstruction refined to a resolution of approximately 2.8 Å (Fig. 1g, Supplementary Figs. 4c, d and Supplementary Table 2). Due to a limited number of twisted-dimer fibrils obtained during data collection, reconstruction of a high-resolution structure was unsuccessful for this fibril type. Therefore, our analysis focused solely on the single protofilament. The fibril exhibits a helical twist of -1.36° and a rise of 4.78 Å (Supplementary Table 2). A structure model was built into the density map showing two separate density regions corresponding to residues C10 to A36 and G57 to N124 (Fig. 1g). The region spanning A37 to H56 was not resolved in the structure, indicating local flexibility.

Each fibril layer adopts a relatively planar conformation and consists of β-stands that show high similarity to ATTRwt fibrils extracted from cardiac tissue (PDB: 8ADE[16]) (Fig. 2a). The ATTRv-F64S fibril contains ten β-stands with β1-β3 located in the N-terminal fragment and β4-β10 in the C-terminal fragment. Hydrophobicity analysis of a single fibril layer revealed three primary hydrophobic grooves formed by residues in (i) the N-terminal fragment (β1-β3),

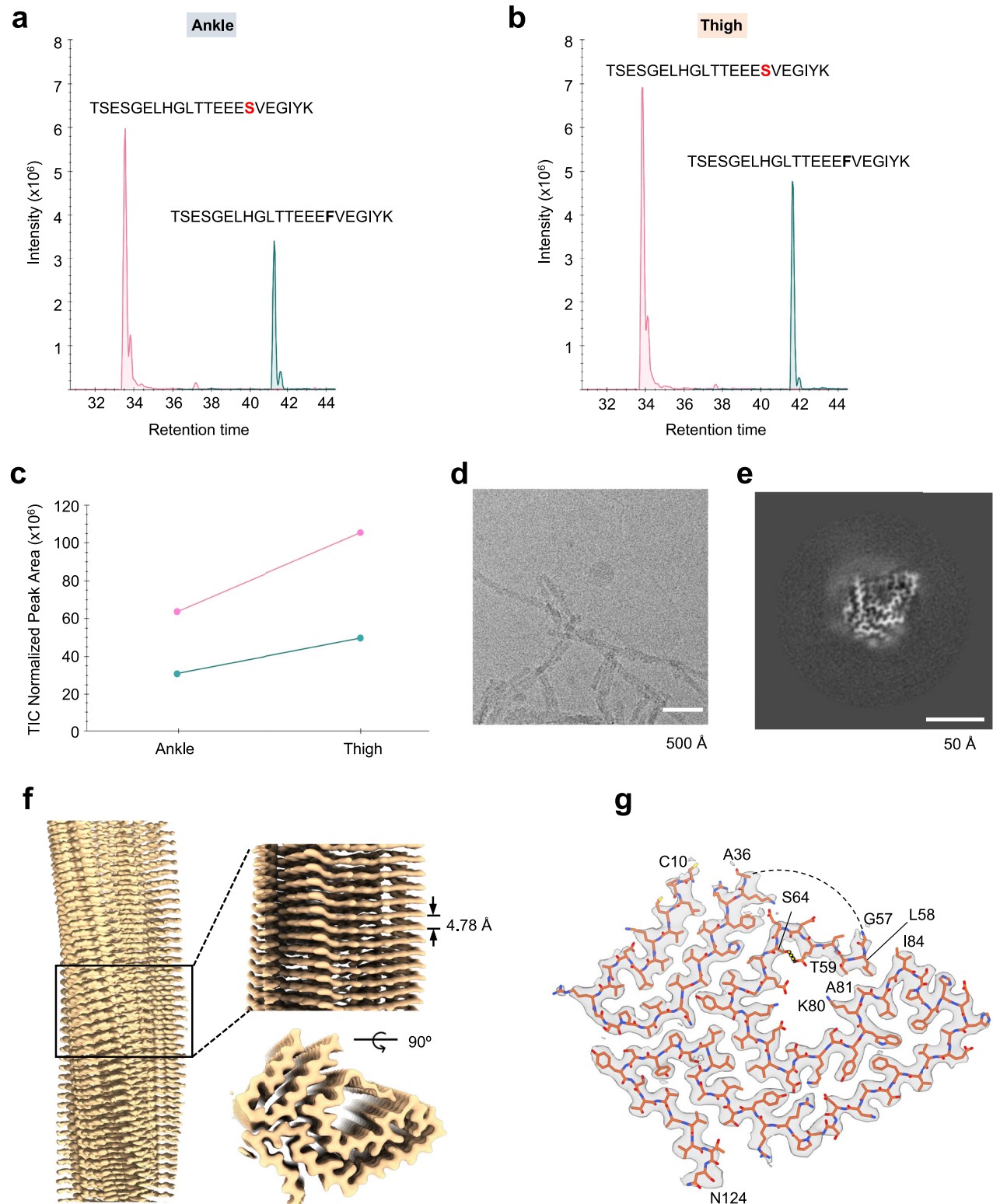

**Fig. 1 | Mass spectrometry analysis and cryo-EM reconstruction of amyloid fibrils from skin tissue of an ATTRv-F64S amyloidosis patient. a,b** Extracted liquid chromatography –mass spectrometry (LC-MS) ion chromatogram of the wild-type (dark green) and F64S-mutant (pink) peptides in fibrils from (**a**) ankle and (**b**) thigh skin tissues. **c** Total ion count (TIC) normalised peak areas indicate a higher intensity of the F64S-mutant peptide in fibrils from both biopsy sites. **d** A representative cryo-EM micrograph of ATTRv-F64S fibrils. A total of 9762 micrographs has been collected. Scale bar, 500 Å. **e** Central slice of the three-dimensional map of ATTRv-F64S fibrils. Scale bar, 50 Å. **f** Cryo-EM density map of the amyloid fibrils. Left, side view of the reconstructed fibril map. Right, close-up views (top and side) of the map with the helical rise indicated. **g** High-resolution electron density map and stick model of a single fibril layer (top view), consisting of an N-terminal fragment (C10 to A36) and a C-terminal fragment (G57 to N124). The disordered region (residues 37–56) of the ATTRv-F64S fibril structure is shown as dashed line. F64S mutation is indicated by an arrow. A hydrogen bond is shown with a yellow dashed line. Source data are provided as a Source Data file.

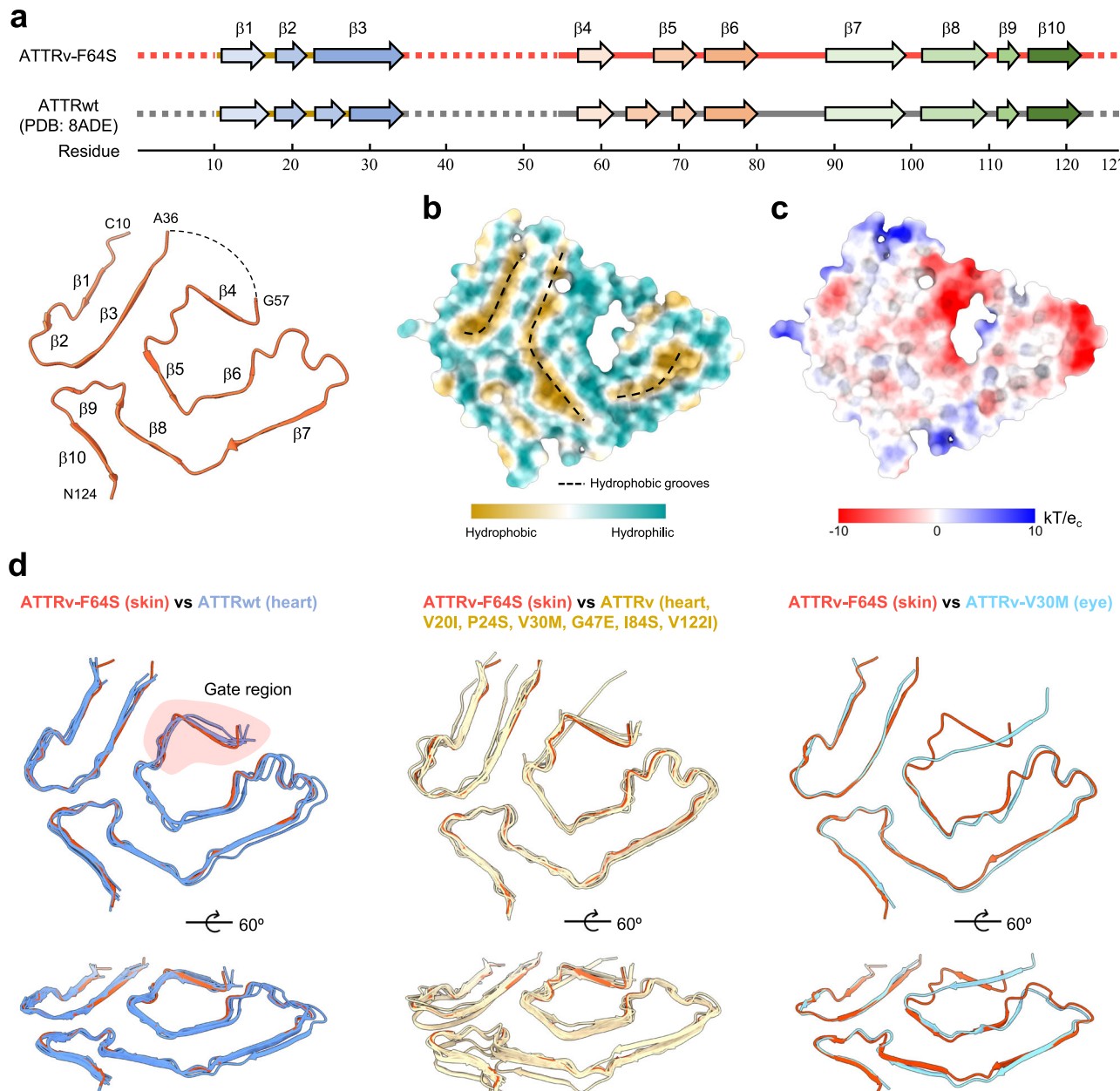

**Fig. 2 | Structural comparison of ATTR fibrils. a** Schematic representation of the secondary structure elements of ATTRv-F64S fibril. Top, ATTRv-F64S fibril β-sheet organization. Bottom, wild-type ATTR (PDB: 8ADE[16]) fibril shown for comparison. Arrows represent β-stands and dashed lines indicate unresolved regions of the fibril protein. Amino acid numbers of the transthyretin (TTR) protein are indicated below. The structure of ATTRv-F64S fibril is shown as a cartoon. **b** Hydrophobicity representation of a single fibril layer (top view). Hydrophobic grooves are shown with dashed lines. **c** Electrostatic surface representation of a single fibril layer (top view). Electrostatic potentials are contoured from -10 (red) to +10 kTe⁻¹ (blue).

Figures are prepared in Chimera X[56]. **d** Structural comparison of ATTR fibrils, including wild-type and variant forms. Left, Views of structural alignment between ATTRv-F64S fibril (tomato red) and ATTRwt fibrils (light blue) extracted from heart (PDB ID: 8ADE[16], 8E7D[18], 8G9R[18], 8GBR[18] and 8E7H[18]). Middle, Views of structural alignment between ATTRv-F64S fibril and cardiac ATTRv fibrils (wheat, PDB ID: 8PKE[17], 8E7I, 6SDZ[14], 8PKF[17], 8E7E[19], 8E7J[19], 8TDN[19], 8TDO[19] and 8PKG[17], including V20I, P24S, V30M, G47E, I84S and V122I mutants.). Right, structural alignment of ATTRv-F64S fibril with ATTRv-V30M fibril (sky blue, PDB ID: 7OB4[15]) from the eye. The gate region is highlighted in pink.

(ii) the C-terminal fragment spanning residues Y78 to A97 (β6- β7) and (iii) the core region involving β3, β5 and β8 (Fig. 2b). Extensive hydrophobic interactions, as well as hydrogen bonding and π-π stacking interactions, contribute to the fibril stability as observed in other ATTR fibrils[18,19]. Electrostatic surface potential analysis indicated two negatively charged patches involving four glutamate residues near β4 and two glutamate residues in a loop between β6 and β7 strands (Fig. 2c). The latter residues were previously shown to be forming the twist-dimer interface of ATTRv-V30M fibrils[15].

**Structural consistency of ATTR fibrils across various tissue types**

Structural alignments were performed on all available ATTR fibril structures in the Protein Data Bank (PDB), including five ATTRwt fibrils, nine ATTRv fibrils derived from cardiac tissue, one ATTRv fibril from the eye and the structure from ankle skin tissue presented in this study (Fig. 2d). To assess structural variability, the backbone displacement was estimated by calculating the root mean square deviation (RMSD) of the Cα atoms ranging from 0.594 to 1.398 Å across the aligned structures (Supplementary Table 3). The structural alignment indicated that the overall fold is conserved throughout all ATTR fibrils

extracted from different tissues, despite local variations around the gate region (Fig. 2d). In all ATTRwt fibrils from cardiac tissue and most ATTRv fibrils, the gate conformation of the ATTRv-F64S fibril is in a closed state. Exceptions are ATTRv-I84S fibrils exhibiting four different conformations (absent, broken, closed and open), and ATTRv-V30M fibrils from the human eye in a blocking state[15,19]. Interestingly, the F64S mutation, which is located within the gate region of the fibril structure, appears to have no effect on the gate conformation.

In sum, our findings suggest that ATTRv-F64S fibrils derived from skin tissue are nearly identical to other ATTR fibrils across different tissues and different genetic variants. This work presents an amyloid fibril structure that has been determined from a skin biopsy of a patient, and we therefore anticipate that this work will guide the development of strain-specific therapeutic strategies for ATTR amyloidosis.

## Discussion

Recent evidence supports a high diagnostic accuracy of skin biopsies for ATTRv even in early and presymptomatic stages of the disease and intraepidermal nerve fibre density (IENFD) correlates with clinical findings of neuropathy[30]. Therefore, we aimed to characterize ATTR fibrils extracted from the tissue of a patient with ATTRv-F64S polyneuropathy. We extracted ATTRv amyloid fibrils from ankle and thigh tissue obtained by skin biopsy. We found that fibrils were more abundant in the ankle tissue than in the thigh tissue in the examined tissue material, suggesting variations in ATTR fibril deposition throughout the body. Using bottom-up proteomics, we identified wild-type and F64S-mutated TTR contributing to the fibril's composition, with the mutant form being detected at a higher proportion. Our analysis thus suggests that the mutant form of TTR may promote fibril formation. We also identify fourteen PTMs on nine sites, ten of which have not been previously described. Some modifications, such as T49 phosphorylation and K70 acetylation, have only been detected in mutant TTR and may be involved in modulating fibril formation (Supplementary Table 1). We mapped these modifications onto the tetrameric and filament structures of transthyretin (Supplementary Fig. 5).

Despite differences in the overall amyloid load between ankle and thigh biopsies, comparable spectral counts were generally detected at most shared PTMs sites, with the exception of K15 acetylation, for which nearly twice as many spectra were measured in the ankle tissue (Supplementary Table 1). While a direct comparison of protein or peptide intensities was limited by different amounts of starting material, maxLFQ normalisation (which accounts for global signal differences) nevertheless suggested a relative enrichment of K15 acetylation in the ankle tissue. Importantly, lysine acetylation has also been shown to modulate amyloidogenicity in other fibrillar proteins. For example, tau acetylation facilitates tau aggregation[31–34] and impairs the degradation of phosphorylated tau[35]. Lysine acetylation of the transactive response DNA binding protein 43 (or TDP-43), promotes its pathological aggregation by impairing nuclear import, RNA binding, and enhancing aberrant phase separation[36,37]. In mutant huntingtin (Htt), acetylation of the N-terminal methionine promotes aggregation[38], whereas phosphorylation of threonine 3 inhibits the aggregation of Htt. The inhibitory effect of T3 phosphorylation on Htt aggregation is reversed by K6 acetylation[12]. Moreover, acetylation of K444 in huntingtin plays a role in neuronal protection by facilitating the clearance of mutant protein[39]. Lastly, acetylation of lysine residues in α-synuclein such as K6, K10, K12 and K80 significantly reduce α-synuclein aggregation and its cytotoxicity, both in vitro and in cells[40,41]. Taken together, the effect of acetylation on amyloid fibril formation is site- and protein-specific. In the case of ATTR, our MS data imply a potential link between K15 acetylation and increased amyloid deposition in ankle tissue. One potential explanation could be that K15 acetylation blocks ubiquitin-mediated degradation of ATTR, thereby promoting fibril deposition. K15 ubiquitination was indeed detected in our LC-MS/MS data of isolated ATTRv-F64S fibrils (Supplementary Table 1). An alternative explanation is that K15 is located at the thyroxine-binding interface, and K15 acetylation might destabilise the native tetrameric fold, usually stabilized by thyroxin binding. Not surprisingly, this interface has been targeted with small molecules such as tafamidis to kinetically stabilize the tetrameric form of TTR and to prevent amyloid formation[42]. However, future experimental validation is needed to uncover the role of K15 acetylation in ATTR amyloidosis.

Previous structural work suggested that mutations destabilizing the monomeric TTR fold drive ATTR fibril formation. The crystal structure of dimeric TTR (PDB: 4TLT[43]) reveals that F64 engages in extensive hydrophobic interactions (Supplementary Fig. 4e). Therefore, mutations of F64 to S, V, I and L – all variants identified in the *TTR* gene[1] –likely destabilize the monomeric TTR fold. Using FoldX to calculate the change in Gibbs free energy (ΔG) for specific mutants show that all described mutants have a positive ΔG, indicating that these mutations indeed destabilize the wild-type fold (Supplementary Fig. 4e)[44]. Notably, F64S has the highest change in ΔG.

While in our study filaments were extracted from thigh and ankle tissue, structural analysis was restricted to fibrils extracted from ankle tissue, due to the low number of fibrils obtained from the thigh tissue. Ankle-derived filaments revealed two types of ATTRv-F64S fibrils: a predominant single protofilament and a low-abundance twisted dimer (Supplementary Figs. 4a and b). Similar variations in fibril morphology have been reported before in ATTRv-V40I and ATTRv-V122Δ cardiac fibrils, as well as in ATTRv-V30M fibrils from the eye[15,22,24]. Other recent cryo-EM studies of ATTR fibrils (wild-type or other genetic variants) from multiple tissues revealed homogeneous fibril morphologies comprising a single protofilament[14,16–21]. However, the presence of additional fibril types at extremely low concentrations cannot be excluded. While the overall morphology is highly conserved in ATTR fibrils, local variations have been observed. The largest local variations occur in the so-called gate region (residues G57–G67), with five distinct gate states, the closed gate being most common (Fig. 2d). Structural polymorphism can commonly be observed in amyloid disorders, such as light-chain (AL) and AA amyloidosis[24,45,46].

What drives (local) structural polymorphisms of ATTR filaments is not yet fully understood, but amyloid structures from different genetic transthyretin variants provide initial insights. For example, the I84S mutation likely destabilises a closed gate conformation by disrupting the hydrophobic interface of residues L58, A81 and I84, thereby promoting alternative conformations (Fig. 1g)[19]. The F64S mutation described in this study is located in the gate region and possibly stabilises a closed gate via a hydrogen bond between the side chains of S64 and E61 (Fig. 1g). However, most ATTR fibril variants and wild-type structures share this conformation suggesting that a closed gate conformation is an intrinsic feature of ATTR fibrils. A structural outlier is the blocking gate conformation observed in ATTRv-V30M fibrils extracted from the vitreous humor of the eye (Fig. 2d)[15]. Moreover, PTMs are known to be critical factors that drive polymorphism of amyloid fibrils. Tau ubiquitination for example influences inter-protofilament packing and contributes to the structural diversity of tau fibrils[47]. Phosphorylation and N-terminal acetylation of α-synuclein can rearrange the amyloid fibril structure[48,49]. The role of PTMs in ATTR fibril formation remains largely unexplored.

Here, we show that K80 (found in a polar cavity) can be acetylated (Supplementary Fig. 5a and Supplementary Table 1), thereby possibly stabilizing the closed gate state by forming a hydrogen bond with T59 (Fig. 1g). In contrast, the blocking gate conformation is likely disfavoured by K80 acetylation because of potential clashes with E62[15]. If K80 acetylation is common, it might explain why the closed gate conformation is frequently observed in ATTR fibrils. In addition, K80 ubiquitination appears to be compatible only with the absent gate

state, in which the resulting polar cavity is large enough to accommodate the ubiquitin chain. We also detected phosphorylation of Y116 (Supplementary Fig. 5a and Supplementary Table 1), which is located at the protofilament interface in ATTRv-V122Δ fibrils[22]. This modification may therefore modulate the formation of multiple protofilaments. Other factors, such as the site of TTR protein production (liver or retinal epithelium) and the microenvironment of fibril deposition, do not appear to be critical for fibril polymorphism, except for the V30M genetic variant[14,15,20]. Further, it is unclear if the quantity of deposited fibrils affects the formation of multiple-protofilament species in different tissues. In essence, PTMs are likely critical factors of ATTR fibril polymorphism, however additional systematic structural and proteomic studies of ATTR fibril formation from different tissues, genotypes, and phenotypes are needed.

In this proof-of-concept study we demonstrate that tissue from minimally invasive skin biopsies can be used to characterize fibril composition (such as wild-type to mutant ratios), PTMs, and to determine the three-dimensional structure by cryo-EM. Fibrils derived from living patients offer the possibility to study filament composition and structure across distinct genotypes, phenotypes and disease stages (fibrils from post-mortem tissue likely reflects the end stages of a disease). Moreover, skin biopsy can be performed longitudinally, enabling within patient follow-up studies that reveal how amyloid fibrils may evolve over time and in response to disease-modifying interventions.

Lastly, the lack of detailed structural data on ATTRwt or ATTRv amyloid fibrils hinders our understanding of the disease mechanisms and directly impedes our ability to design novel drugs. Current therapeutic strategies focus on preventing amyloid formation. For example, RNAi-based therapies are already in clinical use and highly effective when administered in the early disease phases[50]. Similarly, small molecular glues that stabilise the native TTR tetramer fold have been developed[42]. However, therapeutic strategies to remove existing amyloid deposits in tissues remain underdeveloped. Patient-derived fibril structures could guide the design of personalised immunotherapies that are more effective and tailored to dominant fibril conformations and epitope profiles. Such therapies could include de novo designed universal binders, such as mini-proteins, peptides or nanobodies that target specific ATTRwt or ATTRv fibrils. Fusing such binders to antibodies that promote the clearance of amyloid deposits via phagocytosis[51] might hold great potential for both diagnostic imaging of amyloid deposits and immunotherapies of ATTR amyloidosis, paving the way towards personalized medicine.

## Methods

### Skin biopsy
A three mm-diameter punch skin biopsy was performed on the distal leg, 10 cm above the lateral malleolus, and on the thigh, 10 cm above the knee, as previously described[23]. The biopsy was conducted from a patient with ATTRv-F64S polyneuropathy, and a disease duration of three years. The skin samples were flash-frozen and stored at -80 °C until further analysis. To quantify amyloid deposits in skin tissue, 50-μm thin sections were stained with Congo red solution[23] and examined under bright-field (BF) microscopy (Zeiss Axio Lab.A1 Microscope, AxioCam ERc 5 s, Oberkochen, Germany). To evaluate small fibre neuropathy in the skin tissue, at least three non-consecutive 50 μm thin tissue sections were stained each location using a primary antibody against the pan axonal marker protein gene product 9.5 (PGP9.5, rabbit, polyclonal, EMD Millipore Corporation, AB5925, dilution 1:1000) and the nuclear stain DAPI (Sigma-Aldrich, Saint Louis USA, dilution 1:5000). Fluorescence imaging was performed using an inverted fluorescence microscope (Nikon Eclipse Ti-E, Tokyo, Japan). PGP9.5-positive fibres crossing the dermal-epidermal junction were counted according to published protocols[52].

### Fibril extraction from the skin tissue
Amyloid fibrils were extracted from human skin tissue using a modified water extraction protocol[24]. In brief, approximately 5-10 mg frozen skin tissues from ankle or thigh biopsy of a living patient carrying a point mutation F64S were thawed at room temperature and subsequently diced. Both skin samples were processed equally in all subsequent steps. Diced tissues from thigh or ankle were separately transferred to 1.5 ml Eppendorf tubes and washed with 0.5 mL Tris-calcium buffer (20 mM Tris, 140 mM NaCl, 2 mM $CaCl_2$, 0.1 % $NaN_3$, pH 8.0). The resulting suspensions were homogenized using a Kimble pellet pestle (Sigma-Aldrich) and centrifuged for 5 min at 3100 × g. All centrifugation steps were performed at 4 °C. The washing procedure was repeated five more times. Pellets were resuspended in 1 mL freshly prepared collagenase digestion buffer supplemented with protease inhibitor cocktail tablet (PIC) (cOmplete EDTA-free, Roche Diagnostics) and 5 mg/mL crude collagenase from *Clostridium histolyticum* (Sigma-Aldrich). The suspension was incubated overnight at 37 °C with shaking at 150 rpm. Next, the samples were centrifuged at 3100 × g for 30 min. Pellets were resuspended in 0.2 mL Tris-EDTA buffer (20 mM Tris, 140 mM NaCl, 10 mM EDTA, 0.1% $NaN_3$, pH 8.0), homogenized, and centrifuged for 10 min at 3100 × g. This washing step was repeated four more times. For the extraction of amyloid fibrils, all pellets were resuspended in 0.2 mL of ice-cold water and centrifuged for 10 min at 3100 × g. The fibril-containing supernatant was collected, and the extraction step was repeated four more times. Amyloid fibrils extracted ankle and thigh tissues were concentrated by ultracentrifugation at 100,000 x g for one hour. Final pellets were resuspended in 30 μL ice-cold water and used for cryo-EM and mass spectrometry analyses.

### Negative staining analyses
In an initial step, amyloid fibrils extracted from ankle and thigh tissues were analysed by negative-stain transmission electron microscopy. Briefly, 3 μL of each sample were applied onto 400-mesh copper grids with carbon film (Electron Microscopy Sciences) that had been glow discharged for 20 s. The samples were incubated for one minute, after which excess solution was removed by blotting using Whatman® Filter Paper. Grids were then washed twice with water and subsequently stained with 2% uranyl acetate for one minute, followed by blotting to remove excess stain. Grids were examined using an in-house Thermo Fisher Scientific (TFS) Talos L120C G2 transmission electron microscope.

### Cryo-EM sample preparation and data collection
Skin tissue from ankle biopsy contained significantly more amyloid fibrils than skin tissue from thigh biopsy. Consequently, cryo-EM analyses were limited to fibrils extracted from ankle skin tissue. 4 μL of concentrated fibril extraction were applied onto glow-discharged holey carbon grids (Quantifoil R1.2/1.3, 300 mesh). Grids were front blotted for three to four seconds with an additional movement of 1 mm (95% humidity at 15 °C) before being plunged into liquid ethane using an EM GP2 automatic plunge freezer (Leica). The grids were stored in liquid nitrogen until data collection.

The cryo-EM dataset was acquired using a TFS Titan Krios transmission electron microscope at an accelerating voltage of 300 kV. A total of 9763 movies from two grids were collected on a Falcon 4i direct electron detector (equipped with a Selectris X filter) at a nominal magnification of 165,000 x, resulting in a pixel size of 0.726 Å. Data were collected using EPU (TFS) software, with five images recorded per hole, with a set defocus range of −0.6 and −2.0 μm and a total electron dose of 40 e⁻/Å². Data acquisition was monitored using on-the-fly preprocessing in CryoSPARC v.4.2.1[53].

### Helical reconstruction
Beam-induced motion correction for all movies was performed using RELION's own implementation of the UCSF motioncor2 program[29].

Contrast Transfer Function (CTF) parameters were estimated by Gctf[54]. All further processing steps were carried out using RELION, as described previously for amyloid structure determination[55]. Fibrils were auto-picked with a binarization threshold of -6 using a modified version of the Topaz module[28]. Segments were extracted using a box size of 384 or 512 pixels with an inter-box distance of 14.2 Å. The segments were then downscaled to 96 or 128 pixels, resulting in a downscaled pixel size of 2.904 Å. Segments were subjected to iterative rounds of reference-free two-dimensional (2D) classification using regularization parameter T = 3, and 100 classes to remove junk molecules. Final high-quality classes contained a total of 135,574 segments. After one final round of 2D classification, 15 images were selected to generate an initial model using the relion_helix_inimodel2d program[28] by optimizing crossover distances. The best initial model was used as a reference for three-dimensional (3D) classification of re-extracted segments with a box size of 280 pixels. The first round of 3D classification was performed using four classes and a regularization parameter of T = 40, resulting in 12,633 segments that contributed to a high-resolution reconstruction. Two more rounds of 3D classification were performed on these segments, using a single class and increasing T values (40 and 80), with local optimization of the helical twist and rise parameters. In the final round of 3D classification, β-strands were well separated, and large side chains could be resolved. The model and data were then used for high-resolution gold-standard 3D refinement. Iterative Bayesian polishing and CTF refinement were performed on these segments to improve the resolution further. The final ATTR fibril map converged with a helical rise of 4.78 Å and a helical twist of −1.36°. The final map refined to a resolution of 2.82 Å, using standard post-processing by applying a soft mask consisting of 30% of the box size.

## Model building and refinement
The cryo-EM structure of ATTRv-V30M (PDB: 6SDZ[14]) was used as an initial reference for model building. The initial model was fitted into the cryo-EM map using Chimera X[56], followed by manual and iterative building in Coot[57], before real-space refined using PHENIX[58]. Model validation was performed with MolProbity[59]. The FSC curve between the cryo-EM map and the atomic coordinates was calculated using Mtriage[60]. Structural figures were generated in Chimera X. All relevant statistics are summarized in Supplementary Table 2.

## Liquid chromatography – tandem mass spectrometry (LC-MS/MS)
Protein extraction and enzymatic digestion: 300 ng or 30 ng of extracted fibrils, either from the ankle or the thigh (n = 1 for both sites), were dissolved in 50 μL of 8 M urea and 50 mM ammonium bicarbonate (ABC) by sonication at 4 °C in a water bath (Bioruptor, Diagenode; 15 cycles of 30 s on and 30 s off in high mode). Proteins were reduced with 10 mM dithiothreitol for 20 minutes at room temperature, followed by alkylation with 50 mM iodoacetamide for 30 minutes at room temperature. Protein digestion was performed by adding LysC (Wako Fujifilm, 1:100 w/w) for two hours at room temperature, after which the digestion buffer was diluted to 2 M urea and 50 mM ABC for overnight digestion with trypsin at room temperature (Promega, 1:100 w/w). Digestion was stopped using a solution of 2% acetonitrile (ACN) and 0.3% trifluoroacetic acid (TFA) and samples were cleared afterwards by centrifugation at maximum speed for 5 min. Digested peptides were purified using C18 StageTips[61] and eluted with 80% ACN, 0.5% acetic acid. The elution buffer was removed by vacuum centrifugation and purified peptides were resuspended in 2% ACN, 0.5% acetic acid, 0.1% TFA for single-shot LC-MS/MS measurements. For bulk skin analysis, the skin material from the ankle (n = 1) was first thawed on ice before washed once with PBS buffer to remove any remaining blood. The tissue was then manually homogenized in 250 μL of 4% SDS (100 mM TRIS, pH 7.6), prior to sonication in a water bath at 4 °C (Bioruptor, Diagenode; 15 cycles of 30 s on and 30 s off in high mode). The tissue

was incubated for 10 min at 95 °C (Thermomixer, Eppendorf, 800 rpm). The lysate was alkylated with 50 mM iodoacetamide for 30 min at room temperature. After clearing the lysate by centrifugation (5 min, 4 °C, 20,000 × g), proteins were precipitated overnight at -20 °C in 80% acetone. Proteins were pelleted by centrifugation (20 min, 4 °C, 20,000 × g), washed with 80% acetone, and dried at 40 °C. The protein pellet was then resuspended in 8 M urea and 50 mM ABC by sonication in a water bath. Protein digestion and peptide purification were performed as described for the extracted fibrils.

## LC-MS/MS analysis
Peptides were separated on a nanoElute2 HPLC system (Bruker) coupled to a timsTOF HT mass spectrometer (Bruker) via a nanoelectrospray source (Captive spray source, Bruker). 500 ng of bulk skin peptides or the entire purified fibril digests were resuspended in a mixture of water and 0.1% formic acid and loaded onto an in-house column (inner diameter: 75 μm; length: 25 cm) packed with ReproSil-Pur C18-AQ 1.9 μm resin (Dr. Maisch HPLC GmbH). The column was kept at 50 °C in a column oven, and the samples were eluted over ninety (bulk skin) or sixty minutes (fibrils) using a linear gradient between 2 and 35% ACN/0.1% formic acid at a flow rate of 300 nL/min. The mass spectrometer was operated in a data-dependent (DDA)-PASEF mode with 10 PASEF ramps. For the bulk biopsy analysis, the accumulation and ramp times were set to 100 ms, covering a 100-1700 m/z range and a 0.70–1.50 Vs/cm² mobility range, for an estimated cycle time of 1.17 s. The signal intensity threshold was set to 1200 and precursors were actively excluded for 0.4 min after reaching a target intensity of 14,500. For fibril analysis, the accumulation and ramp times were set at 166 ms, covering a 100–1700 m/z range and a 0.60–1.60 Vs/cm² mobility range, for an estimated cycle time of 1.89 s. The signal intensity threshold was set to 1000 and precursors were actively excluded for 0.4 minutes after reaching a target intensity of 20,000. High-sensitivity detection for low sample amounts was activated. For both analyses, the collision energy was ramped up linearly from 20 eV at 0.60 Vs/cm² to 59 eV at 1.60 Vs/cm². Bruker's default active precursor region filter was used to select precursors with charge up to 5 for fragmentation.

## LC-MS/MS Data analysis
Raw data were analysed using FragPipe 21.1[62,63]. MSFragger 4 was used to search MS/MS spectra against the Human UNIPROT database (June 2019), with an additional entry for the F64S-mutated TTR sequence. Cleavage type was set to "enzymatic" with the "stricttrypsin" cleavage rule. The minimum peptide length was set to seven amino acids, the maximum to 50, and the mass range between 500 and 5000 Da. For the regular search, N-terminal protein acetylation and methionine oxidation were set as variable modifications and cysteine carbamidomethylation as a fixed modification. Precursor and fragment mass tolerance were set to 20 ppm, and a false discovery rate (FDR) of 1% was required at the PSM, ion, peptide, and protein levels. PSMs were rescored with MSBooster using deep learning prediction and validated with Percolator using a minimum probability threshold of 0.5. Label-free quantification was carried out with IonQuant 1.10.12. The same parameters as described before were used for the semispecific N-terminal search, except the cleavage type was changed to "semi_N_term". Four separate searches were performed for the analysis of post-translational modifications by adding the following variable modifications: 1) serine/threonine/tyrosine phosphorylation; 2) lysine/arginine monomethylation and dimethylation, and lysine trimethylation; 3) lysine acetylation; 4) lysine ubiquitination. PTM site localization was enabled by running PTM-Prophet with a minimum probability threshold of 0.5. Results were further filtered based on the localization probability of each PTM and the spectral similarity to predicted spectra, which were set at a minimum of 0.8. Ion chromatogram extraction and peak area quantification were performed using Skyline 23.1.0.455.

## Ethical statement

This study was performed in line with the principles of the Declaration of Helsinki and the study protocol, was approved by the Ticino Cantonal Ethics Committee as a re-use of biological samples that are taken as part of the clinical routine (CE TI 2895), and written informed consent was granted (EOC_M-AFRI-001/A).

## Reporting summary

Further information on research design is available in the Nature Portfolio Reporting Summary linked to this article.

## Data availability

The structural data generated in this study have been deposited to the worldwide Protein Data Bank (wwPDB) and the Electron Microscopy Data Bank (EMDB) with the accession codes 9HYW and EMD-52519, respectively. The mass spectrometry proteomics data have been deposited to the ProteomeXchange Consortium via the PRIDE[64] partner repository with the dataset identifier PXD065685. Unless otherwise stated, all data supporting the results of this study can be found in the article, supplementary, and source data files. Source data are provided with this paper.

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

## Acknowledgements

We would like to express our deep gratitude to the patient providing the biological material needed to conduct this research study. We thank A. Reynaud and Y. Pfister for technical assistance; All group members from the Melli and Boland groups for their input and discussion; The computing department at the University of Geneva for providing the infrastructure to perform cryo-EM analysis; N. Roggli for maintaining computing in the Molecular and Cellular Biology department; C. Bauer, A. Howe, and S. Barrass at the DCI-Geneva (aka as cryoGEnic) and E. Uchikawa, B. Beckert, S. Nazarov, and A. Myasnikov from DCI-Lausanne – all for their excellent support in EM data collection and analysis. K. Muir and R. Loewith for critical reading of the manuscript. We are grateful for the generous funding support of our research project from the Swiss National Science Foundation (SNSF) (TMSGI3_211581), the Fondation Roger de Spoelberch, the Fondation pour la recherche en biologie et médecine (all to A.B.), and an AFRI-EOC Research Support Grant 2023–2024 (to G.M.). Source Data are provided with this paper.

## Author contributions

J.Y. and X.Z. extracted fibrils from skin tissue, prepared cryo-EM samples, and determined the cryo-EM structure. M.P. carried out mass spectrometry analysis. E.V. and S.P. were responsible for processing the skin biopsies and performing the immunohistochemical analysis. G.M. was responsible for the conceptualization of the project. G.M. and A.B. directed the project. J.Y., M.P., A.C., G.M., and A.B. wrote the paper with contributions from all other authors.

## Competing interests

The authors declare no competing interests.
