## [Transparent Peer Review file · Nature Communications]

Structure of ATTRv-F64S fibrils isolated from skin tissue of a living patient

Corresponding Author: Professor Andreas Boland

Version 0:

Reviewer comments:

Reviewer #1

(Remarks to the Author)

In this paper the authors use immunohistochemistry, mass spectrometry, and cryo-EM to describe the molecular composition and the structural characteristics of amyloid fibrils extracted from skin biopsies of a patient with ATTRv-F64S. The fibril structure of this variant has not been determined in the past. Also the novelty of this paper is that the fibrils were extracted from a living patient. They find that this approach for isolating ATTR fibrils is feasible and that the most commonly identified single protofilament adopts a closed gate type fold in these types of fibrils.

It was not surprising to see both WT and mutated fibrils within skin biopsies from this heterozygous patient since both types coexist in ATTRv biopsies (e.g. due to non-specific seeding of sticky amyloid plaques from serum TTR). Also reassuring to see that more of the mutated form (amyloidogenic) was pulled down.

It was interesting to see several PTMs in these fibrils some of which were not shared across the two biopsy sites of the same tissue or that have not been reported before. I have no concerns about the MS workflow and methodology used. Any thoughts on the higher spectral counts (assuming these can be considered as a semi-quantitative measure of abundance) for acetylation at K15 which was shared across two sites but more "abundant" in the ankle which also had more amyloid deposition? That would be worth discussing a bit more considering its potential role in modifying amyloidogenicity in other fibrillar proteins.

It was reassuring to see similar core folding conformation to prior structures described especially since these were fibrils derived from a living donor. Were there any structural differences between the thigh and ankle fibrils? Any thoughts on what drives structural variability across tissues? PTMs? Gate region variability? Host tissue factors (e.g. fibril variability within same mutation and tissue but across patients as in Saelices et al PMID: 38233397) ? May be worth discussing a bit more as these factors may influence organ tropism and toxicity.

Despite the technical innovations (high resolution, small starting sample, live patient, skin tissue, extensive PTM analyses) I am having a hard time appreciating the novelty and to some extent significance of the findings and this can be discussed a bit more.

It is nice to be able to isolate fibrils from tiny surgical specimens but this is not needed for clinical practice (e.g. diagnosis, treatment or prognosis) unless the authors can provide literature linking structural findings to patient outcomes (e.g. response to treatment, organ involvement or clinical severity). The novelty or implications for disease biology of some of these findings may need to be more explicitly mentioned and discussed. For example the concluding statement mentions personalized therapies but the authors could elaborate on various scenarios (e.g. engineering amyloid clearing antibodies based on each patients dominant fibril structure and epitopes).

A prospective study within the same patient would be of significant value especially after current therapies (stabilizers, silencers) and especially focusing on how several aspects of the protein described here (wt to mutant ratios, PTMs, structural variants) evolve over time and what this means for patients. Currently most methods (imaging or invasive organ biopsies are snapshots in time. This method could allow for a more global understanding of fibril biology over time). Some of these ideas can be briefly mentioned to help the reader grasp the significance of these findings.

Reviewer #2

(Remarks to the Author)

The manuscript from Dr. Jun Yu and colleagues, entitled "Structure of ATTRv-F64S fibrils isolated from skin tissue of a living patient" presents the cryo-EM structure of transthyretin amyloid fibrils extracted from the skin tissue of a patient with the genetic mutation Phe64Ser (ATTRv-F64S amyloidosis). In previous studies, the authors have demonstrated that skin biopsy of living patients is sufficient for ATTRv fibril diagnostics. In this study, amyloid fibrils were extracted from frozen skin tissues, which yielded enough fibrillary material for cryo-EM structural reconstruction at 2.9 Å. The structure seems carefully determined. The ATTRv-F64S structure is compared to previously published structures of ATTRv fibrils. Overall, the skin fibrils are similar to the ones identified in heart tissues.

The paper is well written (including the method section), to the point, and presents a ATTRv fibrillary structure from a new tissue, which is of high potential interest. The cryoEM study is well-supported with additional experiments including studies of post-translational modifications and characterizations of fragmentations using mass-spectrometry and NMR. A thorough and interesting comparison with other ATTRv fibrils is presented.

I only have some minor issues:

1. In Table 2 and Figure 4c it is stated that the structure is refined at 2.9 Å. However, in Table 3 and on line 190 it is stated that it is a 2.8 Å structure. Please clarify.
2. Out of curiosity, what is the area of skin you need to remove to get the "5-10 mg of frozen skin tissue" used for purification? Skin have many layers, where do you typically find ATTRv amyloid fibrils?
3. The authors make a case that this is the first structure of ATTRv amyloid fibrils isolated from a living patient. However, the fibrils used to determine the cryo-EM structure of ATTRv-V30M fibrils in the eye (in the vitreous humor) were collected by vitrectomy from the eye of a living patient. Please clarify.

Version 1:

Reviewer comments:

Reviewer #1

(Remarks to the Author)

Thank you for revising the paper. My concerns have been addressed

Reviewer #2

(Remarks to the Author)

TITLE: "Structure of ATTRv-F64S fibrils isolated from skin tissue of a living patient"
ID# Nature Communications manuscript NCOMMS-25-55913-T

Response to Reviewers.

We would like to thank the reviewers for taking the time to evaluate our manuscript, and for their constructive and positive feedback on our work. In line with their suggestions, we have expanded the discussion in the revised version of the manuscript to provide a broader context of our findings on ATTR amyloid formation.

Reviewer 1

In this paper the authors use immunohistochemistry, mass spectrometry, and cryo-EM to describe the molecular composition and the structural characteristics of amyloid fibrils extracted from skin biopsies of a patient with ATTRv-F64S. The fibril structure of this variant has not been determined in the past. Also, the novelty of this paper is that the fibrils were extracted from a living patient. They find that this approach for isolating ATTR fibrils is feasible and that the most commonly identified single protofilament adopts a closed gate type fold in these types of fibrils.

It was not surprising to see both WT and mutated fibrils within skin biopsies from this heterozygous patient since both types coexist in ATTRv biopsies (e.g. due to non-specific seeding of sticky amyloid plaques from serum TTR). Also reassuring to see that more of the mutated form (amyloidogenic) was pulled down.

It was interesting to see several PTMs in these fibrils some of which were not shared across the two biopsy sites of the same tissue or that have not been reported before. I have no concerns about the MS workflow and methodology used. Any thoughts on the higher spectral counts (assuming these can be considered as a semi-quantitative measure of abundance) for acetylation at K15 which was shared across two sites but more “abundant” in the ankle which also had more amyloid deposition? That would be worth discussing a bit more considering its potential role in modifying amyloidogenicity in other fibrillar proteins.

We thank the reviewer for raising this interesting point and have modified our discussion. The discussion now includes the role of lysine acetylation in the tau protein, TDP-43, huntingtin and alpha-synuclein. We also speculate on the potential role of K15 acetylation in ATTR amyloidosis. This is described from page 8, line 247 onwards.

It was reassuring to see similar core folding conformation to prior structures described especially since these were fibrils derived from a living donor. Were there any structural differences between the thigh and ankle fibrils? Any thoughts on what drives structural variability across tissues? PTMs? Gate region variability? Host tissue factors (e.g. fibril variability within same mutation and tissue but across patients as in Saelices et al PMID: 38233397)? May be worth discussing a bit more as these factors may influence organ tropism and toxicity.

We fully agree with the comment that comparing fibrils from the two biopsy sites would be of great scientific value. However, due to the limited number of fibrils extracted from thigh tissue, a meaningful fibril reconstruction from this site was not possible in our hands. To address the referee’s comment on fibril variability across different tissues we have now

included an additional section in the discussion on structural variability of ATTR filaments (page 9 line 286).

Despite the technical innovations (high resolution, small starting sample, live patient, skin tissue, extensive PTM analyses) I am having a hard time appreciating the novelty and to some extent significance of the findings and this can be discussed a bit more.

It is nice to be able to isolate fibrils from tiny surgical specimens, but this is not needed for clinical practice (e.g. diagnosis, treatment or prognosis) unless the authors can provide literature linking structural findings to patient outcomes (e.g. response to treatment, organ involvement or clinical severity). The novelty or implications for disease biology of some of these findings may need to be more explicitly mentioned and discussed. For example, the concluding statement mentions personalized therapies, but the authors could elaborate on various scenarios (e.g. engineering amyloid clearing antibodies based on each patient's dominant fibril structure and epitopes).

It is worth noting here that being able to extract filaments from small skin biopsies of living patients massively increases the pool of genetic variations of ATTR or other neurodegenerative diseases (to be tested!) to be studied by cryoEM – in contrast of using biological material from deceased people. It is also plausible to use this proof-of-concept work to study the progression of ATTR or other amyloid diseases by taking skin tissue sample over several years. In the case of this study the onset of neuropathic symptoms was around age 30, a few years before the skin biopsy. For example, extracting filaments during different stages of the disease and determine of specific fibril types are typical for early or late disease progression could now be evaluated. Using biological material from deceased people will almost inevitably reflect the fibril composition and structure at a late stage of the disease.

In addition, and to adequately address this important point, we have significantly extended our discussion on the clinical impact and significance of our findings in the manuscript.

A prospective study within the same patient would be of significant value especially after current therapies (stabilizers, silencers) and especially focusing on how several aspects of the protein described here (wt to mutant ratios, PTMs, structural variants) evolve over time and what this means for patients. Currently most methods (imaging or invasive organ biopsies are snapshots in time. This method could allow for a more global understanding of fibril biology over time). Some of these ideas can be briefly mentioned to help the reader grasp the significance of these findings.

We are very thankful to the reviewer for this great input! We have now incorporated these ideas in multiple sections of our discussion.

Reviewer 2

The manuscript from Dr. Jun Yu and colleagues, entitled "Structure of ATTRv-F64S fibrils isolated from skin tissue of a living patient" presents the cryo-EM structure of transthyretin amyloid fibrils extracted from the skin tissue of a patient with the genetic mutation Phe64Ser (ATTRv-F64S amyloidosis). In previous studies, the authors have demonstrated that skin biopsy of living patients is sufficient for ATTRv fibril diagnostics. In this study, amyloid fibrils were extracted from frozen skin tissues, which yielded enough fibrillary material for cryo-EM structural reconstruction at 2.9 Å. The structure seems carefully determined. The

ATTRv-F64S structure is compared to previously published structures of ATTRv fibrils. Overall, the skin fibrils are similar to the ones identified in heart tissues.

The paper is well written (including the method section), to the point, and presents an ATTRv fibrillary structure from a new tissue, which is of high potential interest. The cryoEM study is well-supported with additional experiments including studies of post-translational modifications and characterizations of fragmentations using mass-spectrometry and NMR. A thorough and interesting comparison with other ATTRv fibrils is presented.

We thank the reviewer for the positive comments on our manuscript.

I only have some minor issues:

1. In Table 2 and Figure 4c it is stated that the structure is refined at 2.9 Å. However, in Table 3 and on line 190 it is stated that it is a 2.8 Å structure. Please clarify.

The values differ due to the distinct methods used for resolution estimation. The overall resolution of a cryo-EM map is typically obtained by calculating the Fourier shell correlation (FSC) for two half-maps binned in resolution shells. We usually report the overall resolution at FSC=0.143 cut off for cryo-EM structures, as stated in main text and Tables 2 and 3. Once a complete and well refined atomic model is available, a model-to-map resolution can be estimated by analysing the FSC between the experimental and model maps, which may be used to estimate the resolution limit. In the latter case, the resolution is reported at FSC=0.5 cut off as shown in Table 2 and Extended Figure 4c and can vary slightly.

2. Out of curiosity, what is the area of skin you need to remove to get the “5-10 mg of frozen skin tissue” used for purification? Skin have many layers, where do you typically find ATTRv amyloid fibrils?

We used a skin biopsy punch of 3mm diameter and roughly 5mm depth (see Methods page 13, line 413). Amyloid deposits in ATTRv patients are mainly found in subepidermis and deep dermis around autonomic structures like sweat glands and arterioles. We described the different localization and morphology of amyloid deposits in skin of several systemic amyloidosis in a previous paper (PMID: 37849451).

3. The authors make a case that this is the first structure of ATTRv amyloid fibrils isolated from a living patient. However, the fibrils used to determine the cryo-EM structure of ATTRv-V30M fibrils in the eye (in the vitreous humor) were collected by vitrectomy from the eye of a living patient. Please clarify.

We thank the reviewer for raising this point, which indeed had escaped our notice. We modified and corrected the text accordingly. Results page 7 line 228.

The new sentences now reads: “Our work presents the first amyloid fibril structure that has been determined from a skin biopsy of a patient, and we therefore anticipate that this work will guide the development of strain-specific therapeutic strategies for ATTR amyloidosis.”

**TITLE: "Structure of ATTRv-F64S fibrils isolated from skin tissue of a living patient"
ID# Nature Communications manuscript NCOMMS-25-55913-T**

Response to Reviewers.

We would like to thank the reviewers for taking the time to evaluate our manuscript, and for their constructive and positive feedback on our work. In line with their suggestions, we have expanded the discussion in the revised version of the manuscript to provide a broader context of our findings on ATTR amyloid formation.

Reviewer 1

In this paper the authors use immunohistochemistry, mass spectrometry, and cryo-EM to describe the molecular composition and the structural characteristics of amyloid fibrils extracted from skin biopsies of a patient with ATTRv-F64S. The fibril structure of this variant has not been determined in the past. Also, the novelty of this paper is that the fibrils were extracted from a living patient. They find that this approach for isolating ATTR fibrils is feasible and that the most commonly identified single protofilament adopts a closed gate type fold in these types of fibrils.

It was not surprising to see both WT and mutated fibrils within skin biopsies from this heterozygous patient since both types coexist in ATTRv biopsies (e.g. due to non-specific seeding of sticky amyloid plaques from serum TTR). Also reassuring to see that more of the mutated form (amyloidogenic) was pulled down.

It was interesting to see several PTMs in these fibrils some of which were not shared across the two biopsy sites of the same tissue or that have not been reported before. I have no concerns about the MS workflow and methodology used. Any thoughts on the higher spectral counts (assuming these can be considered as a semi-quantitative measure of abundance) for acetylation at K15 which was shared across two sites but more “abundant” in the ankle which also had more amyloid deposition? That would be worth discussing a bit more considering its potential role in modifying amyloidogenicity in other fibrillar proteins.

We thank the reviewer for raising this interesting point and have modified our discussion. The discussion now includes the role of lysine acetylation in the tau protein, TDP-43, huntingtin and alpha-synuclein. We also speculate on the potential role of K15 acetylation in ATTR amyloidosis. This is described from page 8, line 247 onwards.

It was reassuring to see similar core folding conformation to prior structures described especially since these were fibrils derived from a living donor. Were there any structural differences between the thigh and ankle fibrils? Any thoughts on what drives structural variability across tissues? PTMs? Gate region variability? Host tissue factors (e.g. fibril variability within same mutation and tissue but across patients as in Saelices et al PMID: 38233397)? May be worth discussing a bit more as these factors may influence organ tropism and toxicity.

We fully agree with the comment that comparing fibrils from the two biopsy sites would be of great scientific value. However, due to the limited number of fibrils extracted from thigh tissue, a meaningful fibril reconstruction from this site was not possible in our hands. To address the referee’s comment on fibril variability across different tissues we have now

included an additional section in the discussion on structural variability of ATTR filaments (page 9 line 286).

Despite the technical innovations (high resolution, small starting sample, live patient, skin tissue, extensive PTM analyses) I am having a hard time appreciating the novelty and to some extent significance of the findings and this can be discussed a bit more.

It is nice to be able to isolate fibrils from tiny surgical specimens, but this is not needed for clinical practice (e.g. diagnosis, treatment or prognosis) unless the authors can provide literature linking structural findings to patient outcomes (e.g. response to treatment, organ involvement or clinical severity). The novelty or implications for disease biology of some of these findings may need to be more explicitly mentioned and discussed. For example, the concluding statement mentions personalized therapies, but the authors could elaborate on various scenarios (e.g. engineering amyloid clearing antibodies based on each patient's dominant fibril structure and epitopes).

It is worth noting here that being able to extract filaments from small skin biopsies of living patients massively increases the pool of genetic variations of ATTR or other neurodegenerative diseases (to be tested!) to be studied by cryoEM – in contrast of using biological material from deceased people. It is also plausible to use this proof-of-concept work to study the progression of ATTR or other amyloid diseases by taking skin tissue sample over several years. In the case of this study the onset of neuropathic symptoms was around age 30, a few years before the skin biopsy. For example, extracting filaments during different stages of the disease and determine of specific fibril types are typical for early or late disease progression could now be evaluated. Using biological material from deceased people will almost inevitably reflect the fibril composition and structure at a late stage of the disease.

In addition, and to adequately address this important point, we have significantly extended our discussion on the clinical impact and significance of our findings in the manuscript.

A prospective study within the same patient would be of significant value especially after current therapies (stabilizers, silencers) and especially focusing on how several aspects of the protein described here (wt to mutant ratios, PTMs, structural variants) evolve over time and what this means for patients. Currently most methods (imaging or invasive organ biopsies are snapshots in time. This method could allow for a more global understanding of fibril biology over time). Some of these ideas can be briefly mentioned to help the reader grasp the significance of these findings.

We are very thankful to the reviewer for this great input! We have now incorporated these ideas in multiple sections of our discussion.

Reviewer 2

The manuscript from Dr. Jun Yu and colleagues, entitled "Structure of ATTRv-F64S fibrils isolated from skin tissue of a living patient" presents the cryo-EM structure of transthyretin amyloid fibrils extracted from the skin tissue of a patient with the genetic mutation Phe64Ser (ATTRv-F64S amyloidosis). In previous studies, the authors have demonstrated that skin biopsy of living patients is sufficient for ATTRv fibril diagnostics. In this study, amyloid fibrils were extracted from frozen skin tissues, which yielded enough fibrillary material for cryo-EM structural reconstruction at 2.9 Å. The structure seems carefully determined. The

ATTRv-F64S structure is compared to previously published structures of ATTRv fibrils. Overall, the skin fibrils are similar to the ones identified in heart tissues.

The paper is well written (including the method section), to the point, and presents an ATTRv fibrillary structure from a new tissue, which is of high potential interest. The cryoEM study is well-supported with additional experiments including studies of post-translational modifications and characterizations of fragmentations using mass-spectrometry and NMR. A thorough and interesting comparison with other ATTRv fibrils is presented.

We thank the reviewer for the positive comments on our manuscript.

I only have some minor issues:

1. In Table 2 and Figure 4c it is stated that the structure is refined at 2.9 Å. However, in Table 3 and on line 190 it is stated that it is a 2.8 Å structure. Please clarify.

The values differ due to the distinct methods used for resolution estimation. The overall resolution of a cryo-EM map is typically obtained by calculating the Fourier shell correlation (FSC) for two half-maps binned in resolution shells. We usually report the overall resolution at FSC=0.143 cut off for cryo-EM structures, as stated in main text and Tables 2 and 3. Once a complete and well refined atomic model is available, a model-to-map resolution can be estimated by analysing the FSC between the experimental and model maps, which may be used to estimate the resolution limit. In the latter case, the resolution is reported at FSC=0.5 cut off as shown in Table 2 and Extended Figure 4c and can vary slightly.

2. Out of curiosity, what is the area of skin you need to remove to get the “5-10 mg of frozen skin tissue” used for purification? Skin have many layers, where do you typically find ATTRv amyloid fibrils?

We used a skin biopsy punch of 3 mm diameter and roughly 5mm depth (see Methods page 13, line 413). Amyloid deposits in ATTRv patients are mainly found in subepidermis and deep dermis around autonomic structures like sweat glands and arterioles. We described the different localization and morphology of amyloid deposits in skin of several systemic amyloidosis in a previous paper (PMID: 37849451).

3. The authors make a case that this is the first structure of ATTRv amyloid fibrils isolated from a living patient. However, the fibrils used to determine the cryo-EM structure of ATTRv-V30M fibrils in the eye (in the vitreous humor) were collected by vitrectomy from the eye of a living patient. Please clarify.

We thank the reviewer for raising this point, which indeed had escaped our notice. We modified and corrected the text accordingly. Results page 7 line 228.

The new sentences now reads: “Our work presents the first amyloid fibril structure that has been determined from a skin biopsy of a patient, and we therefore anticipate that this work will guide the development of strain-specific therapeutic strategies for ATTR amyloidosis.”